# Association between Plasma Omega-3 and Handgrip Strength According to Glycohemoglobin Levels in Older Adults: Results from NHANES 2011–2012

**DOI:** 10.3390/nu14194060

**Published:** 2022-09-29

**Authors:** Raíssa A. B. Batista, Flávia M. S. de Branco, Rafaela Nehme, Erick P. de Oliveira, Geórgia das G. Pena

**Affiliations:** 1Graduate Program in Health Sciences, School of Medicine, Federal University of Uberlandia (UFU), Uberlandia 38402-018, Minas Gerais, Brazil; 2Laboratory of Nutrition, Exercise and Health (LaNES), School of Medicine, Federal University of Uberlandia (UFU), Uberlandia 38402-018, Minas Gerais, Brazil

**Keywords:** oxidative stress, omega-3, older adults, HbA1c, impaired glucose tolerance

## Abstract

Background: Low muscle strength is a predictor of mortality in older adults. Although the evidence concerning hyperglycemia is limited, evidence shows that omega-3 (ω-3) intake may be positively associated with muscle strength. However, the association between plasma ω-3 and muscle strength in older adults according to glycohemoglobin (HbA1c) levels has not yet been investigated. Objective: To evaluate whether plasma ω-3 levels are associated with handgrip strength in individuals over 50 years according to HbA1c levels. Methods: This cross-sectional study included 950 older adults (50–85 years) from NHANES 2011–2012. Linear regression analysis was performed to evaluate the association between plasma ω-3 and handgrip strength in individuals with elevated (≥5.7%) or normal HbA1c levels after adjustments for confounders. Results: Total plasma ω-3, docosahexaenoic acid, eicosapentaenoic acid and alpha-linolenic acid were not associated with handgrip strength in older adults regardless of HbA1c levels. Conclusion: Plasma ω-3 levels are not associated with handgrip strength in individuals over 50 years old independent of HbA1c levels.

## 1. Introduction

Low muscle strength has been associated with the occurrence of falls [1] and higher risk of mortality in older adults [2]. Several components are associated with muscle strength loss, such as hormonal changes [3], physical inactivity [3], inadequate dietary intake [3,4], increased inflammation [5] and oxidative stress [6]. Individuals with hyperglycemia, that can be evaluated through HbA1c levels, may have an accumulation of advanced glycation products (AGES), insulin resistance, increased oxidative stress and inflammation [7]. Thus, several studies have reported that individuals with elevated glucose levels have lower muscle strength when compared with those with normal glucose levels [8,9,10,11]. Therefore, it is important to evaluate whether protective factors for muscle strength have different associations separating the individuals with normal or elevated HbA1c levels.

Ômega-3 (ω-3) intake has been considered a promising protective factor for muscle strength in several populations [12,13,14,15,16,17,18,19,20]. It has been proposed that ω-3 may increase acetylcholine sensitivity and membrane fluidity [21], which can make synaptic transmission faster at the neuromuscular junction and thus resulting in faster contractility [21]. In addition, since increased inflammation seems to predict muscle strength loss in older adults and also in individuals with hyperglycemia [7], it is possible to speculate that individuals ingesting more ω-3 could have higher strength as ω-3 has anti-inflammatory properties [18,20,22,23]. However, the current evidence as to whether ω-3 is positively associated with muscle strength in older adults is conflicting [13,16,19,24] and this evidence is unknown in individuals presenting hyperglycemia.

Few studies have evaluated the association between ω-3 intake and muscle strength in older adults [16,19,24] and, to date, no study has evaluated this association in older adults with hyperglycemia. Robinson et al. (2008) showed that the consumption of fatty fish, a ω-3 food source, was the largest predictor of handgrip strength in older adults [16]. However, this study did not evaluate ω-3 intake per se [16], which limits the conclusion of a direct association with strength. Reinders et al. (2015) investigated the cross-sectional association of total plasma ω-3, alpha-linolenic acid (ALA), docosahexaenoic acid (DHA), and eicosapentaenoic acid (EPA) with muscle strength in older adults and no associations were observed. Rossato et al. (2020a) showed that the intake of total ω-3 (but not EPA, DHA and ALA) was positively associated with muscle strength only in older men, but not in women. Therefore, these studies suggest that ω-3 intake might be associated with muscle strength in older adults, but this conclusion is still conflicting and based on limited data [24]. Although no study has directly evaluated the association between plasma ω-3 and strength in older adults with hyperglycemia, a study showed that reduced dietary ω-3 intake was associated with sarcopenia in older adults, which suggests that ω-3 could be likely associated with muscle strength in this population. However, since no association was performed between ω-3 intake and muscle strength in an isolated form [24], more studies are needed.

Thus, the aim of the present study was to evaluate whether plasma ω-3 levels are associated with handgrip strength in individuals over 50 years according to HbA1c levels (elevated or normal values). We hypothesized that plasma ω-3 would be positively associated with strength in older adults with normal glucose levels, and this association would be stronger in individuals with hyperglycemia.

## 2. Methods

### 2.1. Data Source

This study was based on population data of the older adults aged 50 to 85 years included in the National Health and Nutrition Survey (NHANES) 2011–2012. The NHANES is a national research program coordinated by the National Center for Health Statistics that conducts a population-based, representative survey with data on the health and nutritional status of non-institutionalized individuals in the United States, tracking changes over the years since the 1960. Each year a survey is carried out with approximately 5000 people, collecting demographic, socioeconomic, dietary and health related data and physical and laboratory examinations. The survey is carried out by a complex of multiple stages and stratified sampling. Furthermore, all protocols and research data are publicly available and online, and all participants consented to participate with approval from the National Center for Health Statistics Research Ethics Review Board (NCHS ERB) (Protocol #2011-17 for NHANES cycle 2011–2012) [25].

### 2.2. Sample Selection

This cross-sectional study included all older adults aged 50 to 85 years enrolled in NHANES 2011–2012, who had complete data on plasma ω-3 levels, plasma HbA1c levels, and handgrip strength assessment. Individuals under 50 years of age, missing data from dietary analysis, and individuals who did not have plasma ω-3, anthropometric data (weight and height), and analysis of handgrip strength were excluded from the sample in this study (Figure 1). A total of 950 individuals (476 men and 474 women) were evaluated.

### 2.3. Plasma ω-3 Levels

Plasma fatty acid analysis of the NHANES 2011–2012 database was performed on a fasting subsample. For analysis of fatty acids, a sample of 0.5 mL of plasma and a volume of 100 μL of plasma were used. The ω-3 fatty acids were analyzed by means of gas chromatography—mass spectrometry and expressed in μmol/L. Alpha-linolenic acid (ALA; 18:3n-3), eicosapentaenoic acid (EPA; 20:5n-3) and docosahexaenoic acid (DHA; 22:6n-3) were evaluated in the present study. Ω-3 fatty acids were analyzed separately because they are acquired from different food sources and food consumption is crucial to plasmatic levels [26]. The percentages of ω-3 fatty acids were described in Appendix A. The total plasma ω-3 was established by the sum of ALA, EPA and DHA since they have greater biological importance. The detailed process of fatty acid profile analysis is available in the NHANES manual. In addition to ω-3, plasma linoleic acid was also evaluated. We only included linoleic acid because the other omega-6 fatty acids had incomplete data considering the n-sample used in the present study.

### 2.4. Plasma HbA1c Levels

For measurement of HbA1c in NHANES 2011–2012, the whole blood sample was diluted with a hemolysis solution by the analyzer. Subsequently, a small volume of the treated sample was placed on the analytical column of High Performance Liquid Chromatography (HPLC). Separation was performed using differences in ionic interactions between the cation exchange group on the column resin surface and the hemoglobin components. To extract the hemoglobin fraction (A1c) from the column material, step elution was used. After separation, the hemoglobin fraction passed through the photometer’s flow cell and the analyzer measured the changes in absorbance at 415 nm. The analyzer integrated the raw data and calculated the percentage of the hemoglobin fraction. The entire analysis process took three minutes.

### 2.5. Handgrip Strength

The muscle strength component was obtained through the sum of handgrip strength (Takei Digital Grip Strength Dynamometer, Model T.K.K.5401) from both hands. According to the procedure detailed in the NHANES Muscle Strength—Grip Test data, participants who were able to hold the dynamometer in one hand were tested, but were excluded from our analysis as they did not have the sum of strength. To perform the measurement, the participant was asked to stand (unless the participant was physically limited) and squeeze the dynamometer as hard as possible with one of the dominant or non-dominant hands, exhaling while squeezing to avoid an increase in intrathoracic pressure. The measurement was performed after the trained examiner had explained and demonstrated the procedure to the participant, adjusted the grip size of the dynamometer to the participant’s hand size, and the participant had made a practical trial. Each hand was tested three times, alternately and with a 60 s rest between measurements on the same hand. At the end, the combined handgrip strength was used, calculated by the sum of the highest reading of each hand in kilograms.

### 2.6. Other Variables of Interest

We analyzed other variables of interest to understand the population characteristics and/or to perform the necessary adjustments to the database according the conceptual model. These variables were: sociodemographic data (age, ethnicity, sex, marital status, annual household income, educational level), health conditions and habits (hypertension, pre-diabetes, diabetes, arthritis and smoking), level of physical activity (moderate and vigorous), anthropometric measures (weight, height, body mass index), laboratory variables (fasting glucose level), dietary data (energy, carbohydrate, protein, lipids, saturated fat, monounsaturated fat, polyunsaturated fat, total ω-3, linoleic acid, fiber and alcohol),and medications (insulin and oral hypoglycemic agents).

Sociodemographic data, health conditions and habits, and medication use were collected through the application of questionnaires at home administered by trained interviewers. The determination of the presence of diabetes or hypertension was based on the question: “Have you ever been told by a doctor or other health professional that you have diabetes/high pressure?”. The level of physical activity was established using the Global Physical Activity Questionnaire (GPAQ), which includes 16 questions related to daily activities, leisure activities and sedentary lifestyle, with the metabolic equivalent (MET) score being performed. Through this questionnaire, physical activity was evaluated considering the last 7 days and classified as moderate (activities that take more than 10 min without interruption that require moderate physical effort and cause small increases in breathing or heart rate) and vigorous (activities that take longer than 10 min without interruption that require great physical effort and cause large increases in breathing or heart rate) [27]. The variable physical activity (yes or no) was created based on the performance of any type of physical activity (moderate or vigorous). Anthropometric, strength measurement and laboratory data were performed based on the assessment protocol established for each NHANES item. Dietary data were collected through two 24 h food recalls, the first being in person and the second by telephone.

### 2.7. Statistical Analyses

The sample was characterized according to the HbA1c cutoff points proposed by American Diabetes Association (ADA), which considers normal as <5.7% and elevated as ≥5.7% [28]. We analyzed the same characteristics by tertile of plasma ω-3 in both groups (HbA1c < 5.7% and ≥5.7%). In both characterization analyses, we used linear regression for continuous dependent variables, and logistic regression for categorical dependent variables. To compare plasma ω-3 levels and handgrip strength into these groups, we used linear regression, which allowed us to identify the potential of the association. This linear regression analysis was performed without (Model 1) and with adjustments for potential confounders (Model 2), such as age, race, sex, marital status, annual family income, educational level, smoking (yes, no), arthritis (yes, no), physical activity (yes, no), body mass index (BMI), intake of energy (kcal/day), protein (g/day) and alcohol (g/day). The determination of the variables used in the adjusted model was performed considering the biological plausibility of possible interferences with handgrip strength.

All analyses were performed in the Stata 14 software (StataCorp^®^, College Station, TX, USA) using the “SVY” commands to include information on the sample weight of ‘2-year fatty acid subsample’, sampling units and strata. The results of the analyses were shown as mean or percentage ± standard deviation or confidence interval. Significant difference was considered for a *p*-value < 0.05.

## 3. Results

The characteristics of the population are shown in Table 1. Compared to individuals with normal HbA1c levels, the individuals with elevated HbA1c were older, presented higher glycemic levels, body weight and body mass index, but lower educational level and handgrip strength. A lower prevalence of non-hispanic white individuals and vigorous physical activity levels was observed in this group. The group with elevated HbA1c also had a higher prevalence of smokers, hypertension and diabetes, in addition to a higher insulin use. Regarding dietary intake, these individuals ingested less energy, carbohydrate, protein (g/kg), total ω-3, ALA and fiber.

Population characteristics according to total plasma ω-3 tertiles for individuals with normal and elevated HbA1c are presented in Table 2. Regarding the group with normal HbA1c, the individuals with higher plasma ω-3 presented higher age, fasting blood glucose, higher prevalence of vigorous physical activity, and lower handgrip strength, height, energy and total fat, i.e., saturated and monounsaturated fats intakes. Regarding the group with elevated HbA1c, the individuals with higher plasma ω-3 presented a lower prevalence of men and smokers, whereas the prevalence of moderate physical activity was higher. The plasma fatty acids were higher according to ω-3 plasma tertiles in both groups, as expected.

Linear regression analyses between plasma ω-3 levels and the handgrip strength in individuals with normal and elevated levels of HbA1c are shown in Table 3. In individuals with normal HbA1c levels, total plasma ω-3 and DHA were inversely associated with muscle strength in the unadjusted analysis (Model 1); however, these associations were no longer significant after adjustments for confounders (Model 2). For individuals with elevated HbA1c, only DHA was inversely associated with muscle strength in the unadjusted model (Model 1); however, this association was no longer significant after adjustments for confounders (Model 2). Therefore, total plasma ω-3, EPA, DHA and ALA were not associated with muscle strength both in individuals with normal and elevated HbA1c levels. In addition, in an analysis with the inclusion of adjustment for hypertension, there were no differences in the results found.

## 4. Discussion

The present study showed that plasma ω-3 levels were not associated with handgrip strength in individuals over 50 years old independent of HbA1c levels. To the best of our knowledge, this is the first study that evaluated the association between muscle strength and plasma ω-3 in older adults with elevated glycemia.

The result of the present study is partially in agreement with the literature as the association between ω-3 intake and muscle strength is controversial [13,16,19,24]. Reinders et al. (2015) showed that total plasma ω-3, ALA, DHA and EPA were not associated with muscle strength in older adults, which is in agreement with our results. However, Rossato et al. (2020a) recently showed that the intake of total ω-3 was positively associated with muscle strength in older men. The discrepancy in the results of Rossato et al. (2020a) and the present study may be possibly explained due to differences in population characteristics, as well as in methods to evaluate ω-3 intake and muscle strength. For example, we evaluated older adults according to the HbA1c levels, in which ω-3 intake was estimated by plasma levels of ω-3, and the strength was evaluated by handgrip strength, whereas Rossato et al. (2020a) evaluated general older adults, in which ω-3 intake was evaluated by dietary recall and the strength by Kinetic Communicator Isokinetic Dynamometer. Therefore, all these differences can possibly explain the discrepant results, which shows that more studies are needed to elucidate whether ω-3 intake is associated with muscle strength, not only in individuals with elevated HbA1c levels, but also in other populations.

We hypothesized that plasma ω-3 would be positively associated with strength in older adults with normal glucose levels, and this association would be stronger in individuals with hyperglycemia. Since individuals with hyperglycemia may have lower muscle strength [8,9,10,11] due to increased levels of AGEs, oxidative stress and inflammation [7], we hypothesized that individuals ingesting more ω-3 would have more muscle strength as ω-3 has anti-inflammatory properties [18,20,22,23]. However, contrary to our hypothesis, no association was observed between plasma ω-3 and muscle strength when individuals with elevated HbA1c levels were evaluated. To the best of our knowledge, there is only one study that is (partially) comparable with ours. Okamura et al. (2020) showed that ω-3 intake was negatively associated with the presence of sarcopenia (low muscle strength and mass) in older adults with type 2 diabetes, which is contrary to our results. However, there are important differences between the studies that can explain the discrepant findings. We evaluated the association of linear values of muscle strength with plasma ω-3, which is possibly the real bioavailable value of ω-3 after metabolic conversions [29], whereas Okamura et al. (2020) performed the association of sarcopenia diagnosis (not only strength) with dietary ω-3 evaluated by a brief-type self-administered diet history questionnaire. In addition, different confounders included in the analyses could also have influenced the results of the studies. Therefore, due to limited data and conflicting results, more studies are needed evaluating whether ω-3 intake can be a protective factor for muscle strength for individuals with hyperglycemia.

This study has limitations. This is a cross-sectional study that does not allow for determining a cause–effect relationship. Our data cannot be extrapolated for young adults, individuals with sarcopenia and/or muscle wasting diseases. No adjustments were made for gastrointestinal tract diseases that may influence the metabolism of fatty acids due to the absence of these data in the database. In addition, no adjustments could be made for drugs that may interact with ω-3 metabolism by the difficulty of controlling all the multiple potential interactions between them. However, since we evaluated the plasma ω-3, (i.e., their bioavailable fraction) that were partially metabolized, the absence of these adjustments likely did not influence our results. The analyses are representative of the USA population; therefore, these results are not valid for individuals from other countries. However, this study has several strengths. We evaluated the ω-3 intake using plasma ω-3, which is a blood marker that estimates ω-3 consumption for 2–4 weeks. The glycemic status was evaluated by HbA1c level, which represents the average of glycemia for 2 or 3 months [30]. Since we evaluated a representative sample of USA, the large sample size increased the power to detect associations of small magnitude. In addition, we adjusted our analyses for several important covariates, reducing the risk of bias due to confounding.

We conclude that plasma ω-3 levels are not associated with handgrip strength in individuals over 50 years of age, independent of HbA1c levels.

## Figures and Tables

**Figure 1 nutrients-14-04060-f001:**
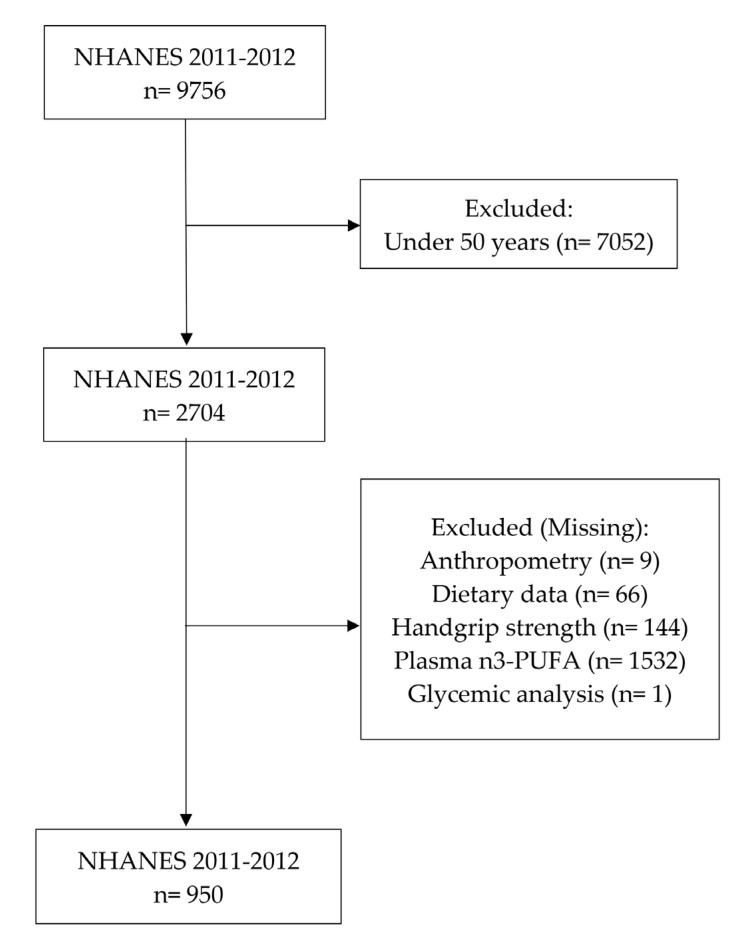
Flowchart of the sample selection from NHANES 2011–2012.

**Table 1 nutrients-14-04060-t001:** Sociodemographic, anthropometric and body composition characteristics in individuals with elevated and normal glycohemoglobin. NHANES, 2011–2012.

Variables	Total	Glycohemoglobin <5.7%	Glycohemoglobin ≥5.7%	*p*-Value
Age, years	63.0 ± 8.9	61.1 ± 7.5	64.8 ± 9.8	**<0.001**
Non-Hispanic white, %	76.5 (68.9–82.6)	84.3 (77.0–89.7)	68.5 (58.5–77.1)	**0.001**
*Sex, %*				0.511
Men	45.8 (41.8–50.0)	47.4 (43.0–52.0)	44.2 (36.5–52.0)	
Women	54.2 (50.0–58.2)	52.6 (48.1–57.0)	56.0 (48.0–63.5)	
*Marital status, %*				0.800
Single/divorced/widowed/never married	31.5 (26.6–36.8)	30.8 (22.9–39.9)	32.2 (26.1–39.0)	
Married/living as married	68.5 (63.2–73.4)	69.2 (60.0–77.1)	67.8 (61.1–74.0)	
*Annual family income, %*				**0.005**
USD 0–19.999	15.9 (12.1–20.7)	11.6 (8.9–14.0)	20.3 (14.1–28.5)	
USD 20.000–54.999	37.5 (30.1–45.6)	32.0 (23.4–42.0)	43.1 (35.3–51.3)	
USD 55.000–74.999	13.7 (9.6–19.3)	14.0 (9.3–19.4)	14.0 (8.2–22.6)	
Over USD 75.000	29.8 (20.6–41.0)	40.4 (30.0–51.8)	19.0 (11.7–29.3)	
Missing	3.0 (1.7–5.4)	2.5 (1.0–6.0)	4.0 (2.0–6.4)	
*Educational level, %*				**<0.001**
High school graduate or under	39.1 (32.4–46.2)	28.0 (21.0–36.2)	50.3 (41.3–59.3)	
Some college or above	60.9 (53.8–67.6)	72.0 (63.8–79.0)	50.0 (40.7–58.7)	
*Health conditions and habits, %*				
Hypertension	51.8 (40.9–55.5)	44.4 (36.6–52.6)	59.3 (48.9–68.9)	**0.008**
Diabetes				**<0.001**
Pre-diabetes	2.9 (1.6–5.1)	1.9 (0.5–6.2)	4.0 (2.0–8.6)	
Yes	15.7 (11.0–21.9)	1.3 (0.5–3.2)	30.3 (22.7–39.2)	
No	81.4 (76.1–85.8)	96.8 (92.8–98.6)	65.8 (58.6–72.4)	
Missing	0.02 (0.00–0.19)	0.04 (0.00–0.38)	-	
Smoking				**0.025**
Yes	16.6 (13.0–21.1)	13.0 (9.6–17.4)	20.3 (14.9–27.0)	
No	83.3 (78.9–87.0)	86.9 (82.6–90.3)	79.7 (73.0–85.1)	
Missing	0.03 (0.00–0.26)	0.07 (0.01–0.55)	-	
Arthritis				0.130
Yes	41.1 (36.0–46.5)	37.0 (29.7–45.0)	45.3 (39.0–51.9)	
No	58.6 (53.3–63.8)	62.5 (55.0–69.8)	54.7 (48.1–61.2)	
Missing	0.3 (0.2–0.4)	0.5 (0.3–0.8)	-	
*Physical activity, %*				**0.035**
Yes	46.9 (40.1–53.9)	53.3 (45.9–60.6)	40.5 (31.1–50.6)	
No	53.1 (46.1–60.0)	46.7 (39.4–54.1)	59.5 (49.4–68.9)	
Moderate PA				0.144
Yes	44.2 (37.4–51.3)	48.7 (40.4–56.9)	39.7 (30.4–49.8)	
No	55.8 (48.7–62.6)	51.3 (43.1–59.6)	60.3 (50.2–69.7)	
Vigorous PA				**0.002**
Yes	11.5 (6.8–18.7)	16.5 (9.5–27.0)	6.4 (3.6–11.0)	
No	88.5 (81.3–93.2)	83.5 (73.0–90.5)	93.6 (89.0–96.4)	
*Anthropometrics*				
Weight, kg	82.7 ± 20.7	78.9 ± 15.0	86.5 ± 25.7	**0.002**
Height, m	1.68 ± 0.1	1.69 ± 0.1	1.67 ± 0.1	**0.040**
Body mass index, kg/m^2^	29.3 ± 6.5	27.6 ± 4.4	31.0 ± 8.2	**<0.001**
*Strength*				
Sum of handgrip strength, kg	65.7 ± 20.6	68.2 ± 18.8	63.1 ± 21.7	**0.005**
*Laboratory data*				
Fasting glucose, mg/dL	110.6 ± 30.0	99.5 ± 8.7	121.8 ± 42.2	**<0.001**
Glycohemoglobin, %	5.9 ± 1.0	5.3 ± 0.2	6.4 ± 1.2	**<0.001**
*Plasma Fatty Acids*				
Total plasma ω-3, µmol/L	363.2 ± 177.2	357.2 ± 158.2	369.3 ± 194.1	0.425
ALA, µmol/L	96.1 ± 55.8	91.3 ± 38.5	101.0 ± 72.3	0.097
EPA, µmol/L	87.7 ± 87.9	91.0 ± 93.9	84.4 ± 71.7	0.465
DHA, µmol/L	179.4 ± 83.5	174.9 ± 71.2	183.9 ± 95.4	0.179
Linoleic acid, µmol/L	3895.8 ± 937.6	3845.8 ± 704.3	3946.5 ± 1169.8	0.204
*Dietary intake*				
Energy, kcal	1960.9 ± 672.1	2038.6 ± 597.9	1882.2 ± 729.5	**0.012**
Carbohydrate, g	237.8 ± 91.4	249.2 ± 85.4	226.3 ± 93.5	**0.001**
Protein, g	77.9 ± 29.7	79.6 ± 27.0	76.3 ± 31.9	0.193
Protein, g/kg	1.0 ± 0.4	1.0 ± 0.3	0.9 ± 0.4	**0.002**
Lipids, g	74.0 ± 32.0	75.9 ± 26.9	72.2 ± 37.0	0.157
Saturated fat, g	23.6 ± 11.2	24.2 ± 9.4	23.0 ± 13.0	0.139
Monounsaturated fat, g	26.6 ± 12.8	27.1 ± 10.8	26.0 ± 14.7	0.292
Polyunsaturated fat, g	17.8 ± 8.7	18.4 ± 7.9	17.2 ± 9.4	0.122
Total ω-3, g	1.77 ± 0.97	1.87 ± 0.94	1.68 ± 0.94	**0.030**
ALA, g	1.69 ± 0.93	1.78 ± 0.91	1.60 ± 0.89	**0.034**
EPA, g	0.03 ± 0.07	0.03 ± 0.05	0.02 ± 0.07	0.113
DHA, g	0.06 ± 0.11	0.06 ± 0.01	0.06 ± 0.12	0.456
Linoleic acid, g	15.7 ± 7.9	16.2 ± 7.1	15.2 ± 8.5	0.148
Fiber, g	18.2 ± 10.4	19.6 ± 10.9	16.7 ± 8.6	**0.001**
Alcohol, g	8.8 ± 20.2	10.9 ± 18.0	6.7 ± 21.9	0.114
*Medicines, %*				
*Oral hypoglycemic agents* Yes No Not eligible *Insulin* Yes No	13.7 (9.0–20.4) 4.8 (3.1–7.4) 81.5 (76.1–85.8) 5.30 (3.55–7.83) 94.70 (92.17–96.45)	0.7 (0.2–2.8) 2.4 (0.9–6.3) 96.9 (92.9–98.6) 0.03 (0.00–0.31) 99.96 (99.69–100.00)	26.9 (19.0–36.4) 7.3 (4.0–13.0) 65.8 (58.6–72.4) 10.62 (7.37–15.08) 89.37 (84.92–92.63)	0.079 **<0.001**

*p* < 0.05 are presented in bold type. Heading variables are presented in bold and italic type in order for better visualization. Notes: DHA: docosahexaenoic acid; EPA: eicosapentaenoic acid; ALA: alpha linolenic acid. Data described as mean (standard deviation) or percentage (confidence interval).

**Table 2 nutrients-14-04060-t002:** Sociodemographic, anthropometric and body composition characteristics in individuals with elevated and normal glycohemoglobin and according to plasma omega-3 tertiles. NHANES, 2011–2012.

Variables	Glycohemoglobin < 5.7% (50.3%)				Glycohemoglobin ≥ 5.7% (49.7%)			
	Tertile 1 96.8–268.8 µmol/L	Tertile 2 270–385.9 µmol/L	Tertile 3 387.6–1803.7 µmol/L	*p*-Trend	Tertile 1 115.1–281.2 µmol/L	Tertile 2 281.6–383.2 µmol/L	Tertile 3 385.4–1561 µmol/L	*p*-Trend
Age, years	59.5 ± 7.6	61.9 ± 8.8	62.2 ± 8.7	0.013	63.8 ± 8.4	65.1 ± 9.4	65.7 ± 8.9	0.149
Non-Hispanic white, %	85.9 (76.1–92.1)	82.0 (69.5–90.1)	85.0 (76.4–90.8)	0.833	73.8 (61.4–83.3)	63.6 (50.5–74.9)	67.5 (57.8–75.9)	0.201
*Sex, %*				0.084				**0.004**
Men	57.8 (50.9–64.5)	37.4 (28.4–47.4)	46.1 (36.4–56.1)		49.2 (37.3–61.3)	54.3 (42.6–65.6)	29.3 (20.7–39.6)	
Women	42.2 (35.5–49.1)	62.6 (52.6–71.6)	53.9 (43.9–63.6)		50.8 (38.7–62.7)	45.7 (34.4–57.4)	70.7 (60.4–79.3)	
*Marital status, %*				0.702				0.139
Single/divorced/widowed/never married	70.2 (54.9–82.0)	71.5 (55.6–83.4)	65.9 (49.4–79.4)		73.1 (65.3–79.6)	63.7 (50.2–75.4)	66.1 (55.9–75.1)	
Married/ living as married	29.8 (18.0–45.1)	28.5 (16.7–44.4)	34.1 (20.6–50.6)		26.9 (20.4–34.7)	36.3 (24.6–49.8)	33.9 (24.9–44.1)	
*Annual family income, %*				0.021				0.217
$0–19.999	15.3 (10.7–21.3)	11.9 (6.0–21.9)	7.3 (3.7–14.1)		22.4 (12.0–38.0)	21.9 (16.0–29.1)	16.6 (11.1–24.1)	
$20.000–54.999	35.5 (25.9–46.4)	29.7 (17.2–46.3)	30.5 (18.6–45.8)		41.6 (31.7–52.3)	42.0 (29.5–55.7)	45.8 (32.3–59.8)	
$55.000–74.999	18.9 (11.6–29.3)	8.3 (3.4–18.9)	13.0 (5.4–28.4)		13.9 (6.3–27.9)	12.1 (6.2–22.5)	15.6 (9.7–24.1)	
Over $75.000	27.6 (17.1–41.4)	48.7 (31.6–66.0)	46.0 (33.6–59.0)		18.1 (9.3–32.5)	19.0 (9.7–33.8)	20.0 (10.2–35.3)	
Missing	2.7 (0.6–11.9)	1.5 (0.6–3.4)	3.2 (07–13.1)		3.9 (1.2–12.4)	5.0 (2.0–12.0)	2.1 (0.9–4.9)	
*Educational level, %*				0.870				0.910
High school graduate or under	25.0 (16.2–36.7)	35.4 (22.7–50.5)	23.8 (15.7–34.4)		48.8 (36.4–61.2)	54.5 (41.8–66.6)	48.1 (36.8–59.5)	
Some college or above	75.0 (63.3–83.8)	64.6 (49.5–77.3)	76.2 (65.6–84.3)		51.2 (38.8–63.6)	45.5 (33.4–58.2)	51.9 (40.5–63.2)	
*Health conditions and habits, %*								
Hypertension	47.6 (33.4–62.2)	42.9 (29.9–56.9)	42.6 (30.0–56.3)	0.509	57.7 (37.8–75.4)	59.4 (49.7–68.5)	60.9 (50.3–70.6)	0.749
Diabetes				0.926				0.144
Pre-diabetes	2.9 (0.5–15.4)	0.7 (0.2–2.3)	2.0 (0.2–16.4)		1.6 (0.5–4.7)	3.0 (0.9–10.2)	7.0 (2.5–18.0)	
Yes	0.7 (0.2–3.5)	1.0 (0.3–3.5)	2.1 (0.5–8.7)		37.7 (28.0–48.4)	26.2 (21.4–31.7)	26.3 (15.2–41.5)	
No	96.3 (85.9–99.1)	98.4 (95.8–99.4)	95.7 (84.9–98.7)		60.7 (49.8–70.7)	70.8 (65.7–75.4)	66.6 (53.6–77.6)	
Missing	0.1 (0.0–1.1)	-	-		-	-	-	
Smoking				0.336				**0.033**
Yes	17.0 (8.9–30.2)	12.0 (6.0–22.4)	9.7 (4.6–19.4)		29.3 (19.4–41.6)	19.7 (11.3–32.1)	11.1(5.3–21.8)	
No	83.0 (69.8–91.1)	87.8 (77.6–93.8)	90.3 (80.6–95.4)		70.7 (58.4–80.6)	80.3 (67.9–88.7)	88.9 (78.2–94.7)	
Missing	-	0.2 (0.0–1.7)	-		-	-	-	
Arthritis				0.430				0.613
Yes	42.3 (26.6–59.8)	34.7 (23.4–48.0)	33.7 (21.4–48.6)		50.4 (36.8–63.9)	38.9 (32.6–45.6)	45.8 (32.9–59.4)	
No	57.7 (40.2–73.4)	63.7 (50.3–75.3)	66.3 (51.4–78.6)		49.6 (36.1–63.2)	61.1 (54.4–67.4)	54.2 (40.6–67.1)	
Missing	-	1.6 (1.0–2.6)	-		-	-	-	
*Physical activity, %*				0.090				**0.026**
Yes	46.0 (32.5–60.1)	52.5 (40.6–64.2)	61.9 (51.2–71.5)		32.3 (22.0–44.5)	40.9 (30.2–52.5)	49.1 (34.7–63.6)	
No	54.0 (40.0–67.5)	47.5 (35.7–59.4)	38.1 (28.5–48.8)		67.7 (55.5–78.0)	59.1 (47.5–69.8)	50.9 (36.4–65.3)	
Moderate PA				0.403				**0.028**
Yes	44.9 (31.7–59.2)	49.7 (36.6–62.8)	51.7 (41.1–62.2)		31.7 (21.6–43.8)	40.0 (29.3–51.7)	48.2 (33.9–62.8)	
No	55.1 (40.8–68.6)	50.3 (37.2–63.4)	48.3 (37.8–58.9)		68.3 (56.2–78.4)	60.0 (48.3–70.7)	51.8 (37.2–66.2)	
Vigorous PA				<0.001				0.659
Yes	6.9 (2.4–18.4)	16.0 (7.7–30.4)	27.3 (17.0–40.7)		4.3 (1.4–12.7)	9.3 (3.8–21.6)	5.9 (1.9–16.6)	
No	93.1 (81.6–97.6)	84.0 (69.6–92.3)	72.7 (59.3–83.0)		95.7 (87.3–98.6)	90.7 (78.4–96.3)	94.1 (83.4–98.1)	
*Anthropometrics*								
Weight, kg	82.2 ± 16.4	77.7 ± 16.5	76.7 ± 17.1	0.050	89.7 ± 27.8	86.9 ± 19.9	82.5 ± 19.3	0.080
Height, m	1.71 ± 0.1	1.66 ± 0.1	1.68 ± 0.1	0.013	1.68 ± 0.1	1.67 ± 0.1	1.65 ± 0.1	**0.017**
Body mass index, kg/m^2^	27.8 ± 4.5	28.1 ± 5.3	27.0 ± 4.9	0.331	31.6 ± 8.9	31.1 ± 6.5	30.3 ± 6.1	0.340
*Strength*								
Sum of handgrip strength, kg	73.6 ± 20.8	63.8 ± 20.4	66.6 ± 21.0	0.019	63.6 ± 19.0	66.8 ± 20.3	59.3 ± 19.1	0.187
*Laboratory data*								
Fasting glucose, mg/dL	97.5 ± 9.0	99.9 ± 10.9	101.3 ± 8.8	0.002	125.1 ± 37.0	116.3 ± 28.2	123.7 ± 46.4	0.795
Glycohemoglobin, %	5.3 ± 0.2	5.4 ± 0.2	5.3 ± 0.3	0.764	6.6 ± 1.2	6.3 ± 1.0	6.4 ± 1.1	0.167
*Plasma Fatty Acids*								
Total plasma ω-3, µmol/L	214.5 ± 39.0	324.2 ± 31.3	543.4 ± 194.2	<0.001	227.3 ± 34.4	334.9 ± 28.6	556.0 ± 188.4	**<0.001**
ALA, µmol/L	66.0 ± 18.6	90.8 ± 26.8	118.9 ± 57.5	<0.001	68.7 ± 23.4	94.0 ± 32.3	142.7 ± 94.3	**<0.001**
EPA, µmol/L	46.2 ± 19.1	66.6 ± 22.9	163.4 ± 161.2	0.001	42.8 ± 14.6	70.3 ± 26.2	143.0 ± 82.1	<0.001
DHA, µmol/L	102.3 ± 27.8	166.8 ± 29.3	261.1 ± 70.2	<0.001	115.8 ± 27.8	170.7 ± 33.5	270.4 ± 92.2	**<0.001**
Linoleic acid, µmol/L	3451.3 ± 549.2	3966.6 ± 789.9	4149.9 ± 851.4	<0.001	3471.5 ± 738.3	3963.1 ± 897.9	4445.5 ± 1272.5	**<0.001**
*Dietary intake*								
Energy, kcal	2190.5 ± 761.8	1946.1 ± 633.5	1967.0 ± 561.1	0.019	1918.9 ± 676.2	1908.6 ± 664.3	1817.3 ± 636.2	0.331
Carbohydrate, g	259.4 ± 99.0	235.8 ± 94.9	251.5 ± 92.1	0.593	225.0 ± 84.9	234.7 ± 91.4	219.8 ± 77.2	0.689
Protein, g	85.9 ± 36.9	76.1 ± 24.8	76.2 ± 25.4	0.076	78.1 ± 30.9	76.8 ± 26.9	73.9 ± 28.1	0.410
Protein, g/kg	1.1 ± 0.4	1.0 ± 0.3	1.0 ± 0.3	0.419	0.9 ± 0.4	0.9 ± 0.3	0.9 ± 0.4	0.985
Lipids, g	85.2 ± 35.4	70.8 ± 27.7	70.9 ± 23.0	0.014	77.9 ± 34.3	71.1 ± 33.9	67.1 ± 31.3	0.106
Saturated fat, g	28.4 ± 11.6	21.6 ± 8.8	22.1 ± 9.3	0.004	24.9 ± 11.7	22.3 ± 11.6	21.6 ± 11.8	0.096
Monounsaturated fat, g	30.7 ± 13.5	25.5 ± 12.8	25.0 ± 8.7	0.010	28.3 ± 13.9	25.9 ± 13.2	23.5 ± 12.3	0.104
Polyunsaturated fat, g	19.4 ± 10.1	17.9 ± 7.5	18.0 ± 8.6	0.397	18.1 ± 8.7	17.2 ± 9.0	16.3 ± 7.8	0.276
Total ω-3, g	1.81 ± 0.96	1.79 ± 0.80	2.01 ± 1.35	0.325	1.72 ± 0.84	1.63 ± 0.86	1.70 ± 0.86	0.911
ALA, g	1.73 ± 0.90	1.74 ± 0.78	1.88 ± 1.33	0.418	1.66 ± 0.80	1.56 ± 0.82	1.58 ± 0.80	0.617
EPA, g	0.03 ± 0.06	0.02 ± 0.03	0.05 ± 0.10	0.078	0.02 ± 0.03	0.02 ± 0.05	0.04 ± 0.10	**0.004**
DHA, g	0.10 ± 0.10	0.04 ± 0.06	0.09 ± 0.15	0.185	0.04 ± 0.06	0.05 ± 0.08	0.08 ± 0.17	**0.022**
Linoleic acid, g	17.1 ± 9.1	15.7 ± 6.8	15.6 ± 7.7	0.299	16.1 ± 7.8	15.2 ± 8.2	14.3 ± 7.0	0.222
Fiber, g	17.2 ± 8.2	19.4 ± 11.6	22.5 ± 15.9	0.062	15.2 ± 7.0	17.8 ± 8.2	17.4 ± 8.1	0.054
Alcohol, g	10.3 ± 15.9	13.9 ± 27.4	8.7 ± 15.6	0.699	3.8 ± 10.8	7.1 ± 16.7	9.5 ± 28.6	0.242
*Medicines, %*								
Oral hypoglycemic agents Yes No Not eligible Insulin No Yes	0.3 (0.0–2.5) 3.3 (0.7–14.5) 96.4 (85.8–99.2) - -	0.7 (0.1–3.3) 1.0 (0.3–2.8) 98.4 (95.8–99.4) 99.88 (99.04–99.98) 0.12 (0.02–1.00)	1.3 (0.2–9.7) 2.9 (0.5–14.2) 95.9 (84.9–99.0) - -	0.884 0.755	31.6 (21.3–44.2) 7.6 (3.6–15.6) 60.7 (49.8–70.7) 85.30 (75.33–91.68) 14.71 (8.32–24.67)	21.8 (17.2–27.3) 7.4 (3.7–14.3) 70.8 (65.7–75.4) 93.09 (87.20–96.40) 6.90 (3.61–12.80)	26.5 (15.0–42.4) 6.9 (2.8–15.6) 66.7 (53.6–77.6) 90.26 (82.51–94.79) 9.74 (5.21–17.49)	0.800 0.110

*p* < 0.05 are presented in bold type. Heading variables are presented in bold and italic type in order for better visualization. Notes: DHA: docosahexaenoic acid; EPA: eicosapentaenoic acid; ALA: alpha linolenic acid. Data described as mean (standard deviation) or percentage (confidence interval).

**Table 3 nutrients-14-04060-t003:** Linear regression of plasma omega-3 levels and handgrip strength in individuals with elevated and normal glycohemoglobin. NHANES, 2011–2012.

	Glycohemoglobin < 5.7%				Glycohemoglobin ≥ 5.7%			
	Model 1		Model 2		Model 1		Model 2	
	β (95% CI)	*p*-Value	β (95% CI)	*p*-Value	β (95% CI)	*p*-Value	β (95% CI)	*p*-Value
Total plasma ω-3, µmol/L	−0.02 (−0.03; −0.00)	0.018	−0.00 (−0.01; 0.01)	0.936	−0.01 (−0.03; 0.01)	0.272	0.00 (−0.00; 0.01)	0.525
ALA, µmol/L	−0.01 (−0.10; 0.08)	0.819	0.01 (−0.02; 0.06)	0.429	0.01 (−0.03; 0.06)	0.614	0.01 (−0.00; 0.03)	0.148
EPA, µmol/L	−0.01 (−0.03; 0.00)	0.094	−0.00 (−0.01; 0.01)	0.895	−0.03 (−0.06; 0.01)	0.105	−0.00 (−0.01; 0.01)	0.415
DHA, µmol/L	−0.05 (−0.07; −0.02)	0.001	−0.01 (−0.02; 0.01)	0.485	−0.03 (−0.05; −0.01)	0.017	0.00 (−0.01; 0.01)	0.750

*p* < 0.05 are presented in bold type. Notes: DHA: docosahexaenoic acid; EPA: eicosapentaenoic acid; ALA: alpha linolenic acid. Model 1: crude analysis; Model 2: adjusted for age, race, sex, marital status, annual family income, educational level, smoking (yes, no), arthritis (yes, no), physical activity (yes, no), body mass index (BMI), energy intake (kcal/day), protein intake (g/day) and alcohol intake (g/day).

## Data Availability

All data and manuals of NHANES 2011-2012 are available at https://wwwn.cdc.gov/nchs/nhanes/.

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
