# Peer review of "Association between Plasma Omega-3 and Handgrip Strength According to Glycohemoglobin Levels in Older Adults: Results from NHANES 2011–2012"

_nutrients, 2022, doi:10.3390/nu14194060_

Round 1
Reviewer 1 Report
Content suggestions:
The study is performed on a high statistical level of evidence with processing relatively a lot of data. However, I would like to kindly ask the authors the following:
-
Do they have the information about the other comorbidities of the included patients except their metabolically associated diseases (e.g. diseases of gastrointestinal tract can influence the metabolism of fatty acids and glycids...) ?
-
Do the authors have the data on all the drugs that may interact with the metabolism of ω-3 (e.g. proton pump inhibitors...) ?
From my perspective, the article can be improved after incorporation of the answers to the questions and after such a minor revision.
Author Response
Uberlândia, Brazil
September 16th, 2022
Dear Reviewer#1,
Yours sincerely,
Geórgia das Graças Pena
Graduate Program in Health Sciences
Federal University of Uberlândia, Uberlândia
Minas Gerais, Brazil
E-mail: georgia@ufu.br
ANSWERS TO THE REVIEWER’S COMMENTS
(The responses are shown immediately below the reviewer’s comments)
Comments from Reviewer #1
Comment 1. Do they have the information about the other comorbidities of the included patients except their metabolically associated diseases (e.g. diseases of gastrointestinal tract can influence the metabolism of fatty acids and glycids...) ?
Comment to Reviewer#1: the first point is that, unfortunately, gastrointestinal tract diseases were not considered in this NHANES 2011-2012, making this approach difficult. Secondly, the questionnaires about comorbidities were made through subjective inquiries, for example, “{Have you/Has spoken} ever been told by a doctor or other health care professional that {you/s/he} had {disease or condition}?”. So, we had not a medical diagnosis. Because of that, we choose to add only arthritis (yes/no) considering the important impact on the outcome. Anyway, we use several sociodemographic, clinical, anthropometrical, and lifestyle adjustments (following the literature). We also believe including all potential comorbidities would not change the results. Finally, we are not sure if the concern of the reviewer was about the complex interactions between gastrointestinal tract diseases and the w-3 consumption. If that is the case, we would highlight that since we used the plasma w-3 data, we are looking at their bioavailable fraction, i.e, after all the possible digestive and abortive interactions. Anyway, to become the text clearer, we add the lack of others comorbidities adjustments as a limitation. Please see the new sentences highlighted in our manuscript (Lines 285-281).
Comment 2. Do the authors have the data on all the drugs that may interact with the metabolism of ω-3 (e.g. proton pump inhibitors...) ?
Comment to Reviewer#1: We recognize these interactions could be frequent since there are at least 72 drugs known for their interaction with fish oils (including proton pump inhibitors such as omeprazole). On the other hand, we know there are many complex interactions drug-drug that could also influence the absorption and biochemical metabolism and we cannot control these variables. Despite these points, we highlighted we evaluated the bioavailable fraction of omega-3 (plasma) and we applied many important adjustments following the literature (also mentioned in comment 1). So, again, since we evaluated the plasma w-3 data, we are looking at their bioavailable fraction, i.e, after all the possible digestive and abortive interactions. Anyway, to become the text clearer, we add the lack of drugs adjustments as a limitation. Please see the new sentences highlighted in our manuscript (Lines 285-281).
Reviewer 2 Report
The manuscript is interesting but there is a some lack of information.
Please provide in the method how the omega-3 fatty acid has calculated?. How the author has quantified?. Whether used internal standards, if it so which internal standard has used?
Please report other fatty acids?
Also provide the percentage of all fatty acids in supplementary.
Author Response
Uberlândia, Brazil
September 16th, 2022
Dear Reviewer#2,
Thank you for the opportunity to revise the manuscript Nutrients-1899712 "Association between plasma omega-3 and handgrip strength according to glycohemoglobin levels in older adults: Results from NHANES 2011-2012".
We hope that you will find our response to your comments satisfactory and suitable for publication. We are willing to finish the revised version of the manuscript, including any further suggestions that you may have.
Yours sincerely,
Geórgia das Graças Pena
Graduate Program in Health Sciences
Federal University of Uberlândia, Uberlândia
Minas Gerais, Brazil
E-mail: georgia@ufu.br
ANSWERS TO THE REVIEWER’S COMMENTS
(The responses are shown immediately below the reviewer’s comments)
Comments from Reviewer 2
Comment 1. Please provide in the method how the omega-3 fatty acid has calculated? How the author has quantified? Whether used internal standards, if it so which internal standard has used?
Comment to Reviewer#2: In the Method section, we described “The ω-3 fatty acids were analyzed by means of gas chromatography-mass spectrometry and expressed in μmol/L. Alpha-linolenic acid (ALA; 18:3n-3), eicosapentaenoic acid (EPA; 20:5n-3), and docosahexaenoic acid (DHA; 22:6n-3) were evaluated in the present study. Ω-3 fatty acids were analyzed separately because they are acquired from different food sources and food consumption is crucial for the plasmatic level [26]. The percentages of ω-3 fatty acids were described in Supplementary Table (Table S1). The total plasma ω-3 was established by the sum of ALA, EPA, and DHA since they have greater biological importance. The detailed process of fatty acid profile analysis is available in the NHANES manual.”
Comment 2. Please report other fatty acids?
We have reported the main ω-3 fatty acids (EPA, DHA and ALA), since they are the most important ω-3. In addition, besides the ω-3, plasma linoleic acid was also evaluated. We only included linoleic acid because the other omega-6 fatty acids had incomplete data considering the n-sample used in the present study. Although other fatty acids are available in NHANES database, we chose to analyze only the omega-3 fatty acids because the current evidence shows that is more likely that omega-3 can be associated with muscle mass or strength when compared with other fatty acids (https://pubmed.ncbi.nlm.nih.gov/30661906/). In addition, a lower number of individuals had the other fatty acids data available in the NHANES database. Therefore, the inclusion of other fatty acids will reduce the number of individuals evaluated, which is a strength of the present study as we evaluated a representative sample of the USA.
Comment 3. Also provide the percentage of all fatty acids in supplementary.
We have included the percentages of ω-3 fatty acids in Supplementary Table (Table S1), as requested by the reviewer (Lines 391-392).